# Stigmasterol Causes Ovarian Cancer Cell Apoptosis by Inducing Endoplasmic Reticulum and Mitochondrial Dysfunction

**DOI:** 10.3390/pharmaceutics12060488

**Published:** 2020-05-28

**Authors:** Hyocheol Bae, Gwonhwa Song, Whasun Lim

**Affiliations:** 1Department of Biotechnology, College of Life Sciences and Biotechnology, Korea University, Seoul 02841, Korea; bhc7@korea.ac.kr; 2Department of Food and Nutrition, College of Science and Technology, Kookmin University, Seoul 02707, Korea

**Keywords:** stigmasterol, ovarian cancer, apoptosis, endoplasmic reticulum, mitochondria, migration

## Abstract

Background: Phytosterols have physiological effects and are used as medicines or food supplements. Stigmasterol has shown anticancer effects against various cancers such as hepatoma, cholangiocarcinoma, gall bladder carcinoma, endometrial adenocarcinoma and skin, gastric, breast, prostate, and cervical cancer. However, there are no reports on stigmasterol’s effects on ovarian cancer. Methods: We investigated the effects of stigmasterol on proapoptotic signals, mitochondrial function, reactive oxygen species production, and the cytosolic and mitochondrial calcium levels in human ovarian cancer cells, to understand the mechanisms underlying the effects of stigmasterol on ovarian cancer cells. We also conducted migration assay to confirm whether that stigmasterol inhibits ovarian cancer cell migration. Results: Stigmasterol inhibited development of human ovarian cancer cells. However, it induced cell apoptosis, ROS production, and calcium overload in ES2 and OV90 cells. In addition, stigmasterol stimulated cell death by activating the ER-mitochondrial axis. We confirmed that stigmasterol suppressed cell migration and angiogenesis genes in human ovarian cancer cells. Conclusions: Our findings suggest that stigmasterol can be used as a new treatment for ovarian cancer.

## 1. Introduction

Over the years, phytosterols have received special attention because of their ability to lower cholesterol and prevent cardiovascular disease [1,2]. In addition, various pharmacological properties of phytosterols have been reported, including anti-inflammatory, antidiabetic, and anticancer properties [3,4]. Stigmasterol is a plant sterol and is a component of the cell membrane that is responsible for its function [5]. Stigmasterol is an unsaturated phytosterol and is found in many vegetables, including legumes, nuts, seeds, herbs, and edible oils. It is also found in unpasteurized milk; however, stigmasterol is weakened by mild heat; thus, it is destroyed during pasteurization at a low temperature [6]. Stigmasterol is used as an intermediate in the biosynthesis of various hormones, including progesterone, androgens, estrogens, and corticoids; it also serves as a precursor to vitamin D_3_ [4,7,8]. Constant consumption of stigmasterol can reduce blood LDL cholesterol by 8%–10%, which lowers the risk of cardiovascular disease [9]. Further, stigmasterol inhibits the accumulation of fatty acids [10]. In plant seeds, stigmasterol promotes cell proliferation, maturation, and germination [11]. Stigmasterol inhibits the survival of human umbilical vein endothelial cells (HUVECs) and iPSC-derived cardiomyocytes [12]. Notably, stigmasterol suppresses the development of various cancers, including hepatoma [13], cholangiocarcinoma [14], gall bladder carcinoma [15], skin cancer [16], gastric cancer [17], breast cancer [18,19], endometrial adenocarcinoma [18], prostate cancer [20], and cervical cancer [19]. 

Among gynecological cancers, ovarian cancer is the fifth-most common cause of death due to difficulty in its diagnosis, high mortality rate, and frequent relapses [21,22]. Therefore, there is a need for the development of new anticancer drugs. In this study, we investigated the effects of stigmasterol on ES2 and OV90 cells. First, we identified whether stigmasterol induces the activity of apoptosis-related factors in ovarian cancer cells. Next, the effect of stigmasterol on cell death, free radical production, and mitochondrial function of ovarian cancer cells was observed. We also investigated the inhibitory effect of stigmasterol on cell proliferation and development of ovarian cancer. The effects of stigmasterol on intracellular signaling pathways were also studied.

## 2. Materials and Methods 

### 2.1. Reagents

Stigmasterol (cat number: S2424) was purchased from Sigma-Aldrich (St Louis, MO, USA). It was diluted in dimethyl sulfoxide (DMSO) before treatment.

### 2.2. Cell Culture

ES2 and OV90 cells were purchased from American Type Culture Collection (ATCC; Manassas, VA, USA). ES2 (type—clear cell carcinoma) and OV90 (type—serous adenocarcinoma) cell lines were grown in McCoy’s 5A medium supplemented with 10% fetal bovine albumin (FBS). Cells were incubated at 37 °C with 5% carbon dioxide (CO_2_). To confirm the effects of stigmasterol, all cells, including those in the control group, were starved with serum-free medium for 24 h prior to treatment.

### 2.3. Western Blotting

The antibodies used in the Western blotting analyses are listed in Table 1. The ovarian cancer cells were cultured on cell culture dishes and incubated with stigmasterol (0, 5, 10, and 20 μg/mL) for 24 h. The cells were rinsed twice with PBS to remove the treatment medium before protein extraction. The cells were lysed, and protein concentration was calculated via the Bradford assay. Equal amounts of each protein were mixed with loading buffer. The protein loading mixture was denatured at 95 °C for 5 min using a heating block. Denatured proteins were separated based on size using sodium dodecyl sulfate–polyacrylamide gel electrophoresis (SDS-PAGE). The separated proteins were transferred onto nitrocellulose membranes for immunoblotting. The membranes were sequentially incubated with primary and secondary antibodies, and chemiluminescence was detected using the ChemiDoc EQ system (Bio-Rad, Hercules, CA, USA). 

### 2.4. Spheroid Assay

ES2 and OV90 cells were dropped onto the cover of a culture dish (3 × 10^3^ cells). Vehicle-treated and stigmasterol-treated cells were cultured for 72 h. Cancer morphology was observed with a DM3000 microscope (Leica, Wetzlar, Germany). Tumor area was calculated using ImageJ software (http://rsb.info.nih.gov/ij/docs/index.html). 3D density was estimated using ReViSP software (https://sourceforge.net/projects/revisp/).

### 2.5. Annexin V and PI Staining

ES2 and OV90 cells grown to 50% confluence in 6-well plates. FBS-starved cells were incubated with stigmasterol (0, 5, 10, and 20 μg/mL) for 48 h. The cells were washed twice with PBS before being treated with 0.25% trypsin-EDTA. The suspended cells were collected via centrifugation and washed twice using PBS to remove the 0.25% trypsin-EDTA. The collected cells were stained with FITC Annexin V (5 μL) and propidium iodide (PI; 5 μL) for 15 min. Fluorescence was detected using a flow cytometer (BD Bioscience, Franklin Lakes, NJ, USA). All experimental results were estimated by comparing them to vehicle-treated controls.

### 2.6. Cell Cycle Assay

ES2 and OV90 cells were grown to 50% confluence in 6-well plates. FBS-starved cells were incubated with stigmasterol (0, 5, 10, and 20 μg/mL) for 48 h. The cells were washed twice to remove the stigmasterol before 0.25% trypsin-EDTA treatment. The suspended cells were collected via centrifugation and washed twice using PBS to remove the remaining 0.25% trypsin-EDTA. The collected cells were incubated with RNase A and PI for 30 min. Fluorescence was detected using a flow cytometer (BD Bioscience). All experimental results were estimated by comparing them to those of the vehicle-treated control cells.

### 2.7. JC-1 Staining

ES2 and OV90 cells were grown to 50% confluence in 6-well plates. FBS-starved cells were incubated with stigmasterol (0, 5, 10, and 20 μg/mL) for 48 h. The cells were washed twice to remove the stigmasterol before trypsinization. The suspended cells were collected via centrifugation and washed twice with PBS to remove the 0.25% trypsin-EDTA. The cells were then incubated with JC-1 (Sigma-Aldrich, St Louis, MO, USA) for 20 min at 37 °C. JC-1-stained cells were washed twice using 1 × JC-1 staining buffer. JC-1 staining was observed using a flow cytometer (BD Bioscience). All experiments results were calculated by comparing them to those of the cells from the control group.

### 2.8. Measurement of Reactive Oxygen Species (ROS) Production

ES2 and OV90 cells were grown in 6-well plates. The cells were detached with 0.25% trypsin-EDTA, and the suspended cells were collected via centrifugation and rinsed twice with PBS. The cells were treated with 10 μM 2′,7′-dichlorofluorescein diacetate (DCFH-DA; Sigma) for 30 min. The stained cells were washed twice with PBS and treated with stigmasterol (0, 5, 10, and 20 μg/mL) for 1 h. After stigmasterol treatment, the cells were rinsed twice with PBS. ROS production was detected using a flow cytometer (BD Bioscience).

### 2.9. Cytosolic Ca^2+^ Level Analysis

ES2 and OV90 cells were grown to 50% confluence in 6-well plates. FBS-starved cells were incubated with stigmasterol (0, 5, 10, and 20 μg/mL) for 48 h. The cells were washed twice to remove the stigmasterol before 0.25% trypsin-EDTA was added to trypsinize the cells. The suspended cells were collected via centrifugation and washed twice with PBS. The cells were incubated with 3 μM fluo-4 acetoxymethyl ester (AM; Invitrogen) for 20 min at 37 °C. The stained cells were rinsed twice with PBS. Fluo-4 intensity was detected using a flow cytometer (BD Bioscience). All experimental results were calculated by comparing them to those of the cells from the control group.

### 2.10. Mitochondrial Ca^2+^ Level Analysis

ES2 and OV90 cells were grown to 50% confluence in 6-well plates. FBS-starved cells were incubated with stigmasterol (0, 5, 10, and 20 μg/mL) for 48 h. The cells were washed twice to remove the stigmasterol before adding 0.25% trypsin-EDTA. The suspended cells were collected via centrifugation and washed twice with PBS to remove any remaining 0.25% trypsin-EDTA. The collected cells were incubated with Rhod-2 for 20 min at 37 °C and subsequently washed twice using PBS. Rhod-2 fluorescence was detected using a flow cytometer (BD Bioscience). All experimental results were calculated by comparing them to those of the cells from the control group.

### 2.11. Proliferation Assay

The proliferation of ovarian cancer cells was observed using ELISA, a BrdU kit (Roche, Basel, Switzerland), and a microplate reader (BioTek, Winooski, VT, USA). Cells at 50% confluence were treated with various concentrations of stigmasterol in a 96-well plate for 48 h. The cells were then incubated with BrdU for an additional 2 h. BrdU was labeled with anti-BrdU-POD for 90 min. The samples in each well were washed three times with PBS, and a color reaction was induced using a chromogenic substrate. All experiments were calculated by comparison of the results to those of the cells from the control group.

### 2.12. Migration Assay

Cell migration was observed using a Transwell migration assay. Ovarian cancer cells were seeded on Transwell inserts and incubated for 12 h with stigmasterol or vehicle. The Transwell membranes were fixed with methanol for 10 min and stained with hematoxylin for 30 min. The membranes were washed and covered using mounting media. Cell migration was observed with a DM3000 microscope (Leica).

### 2.13. Quantitative Real-Time PCR

Gene expression was measured using quantitative RT-PCR using SYBR green, as previously described [23] (see in Table 2). 

### 2.14. Statistical Analysis

The significance of all experiments was identified through the appropriate error terms according to the expectation of the mean square of the error. P-values less than 0.05 were considered significant. Data are shown as the least-square means (LSMs) with standard errors [24]. (*** = *p* < 0.001, ** = *p* < 0.01, and * = *p* < 0.05)

## 3. Results

### 3.1. Induction of Cell Apoptosis and Inhibition of Cell Aggregation by Stigmasterol in ES2 and OV90 Cells

Western blotting showed that stigmasterol activated proapoptotic signals in ES2 and OV90 cells. Stigmasterol (0, 5, 10, and 20 μg/mL) stimulated cleavage of caspase 3 and caspase 9 in a dose-dependent manner in each cell type. In addition, stigmasterol activated cytochrome c, BAK, and BAX in both cell types. The levels of alpha-tubulin (TUBA) did not show changes following stigmasterol treatment (Figure 1A,B). Stigmasterol increased the tumor area by 150.9% and 146.9% in the case of ES2 and OV90 cells, respectively. However, tumor volume was reduced by 72.8% and 60.1% in ES2 and OV90 cells, respectively, by administration of stigmasterol (20 μg/mL). We identified that ovarian cancer cells cannot aggregate in the presence of stigmasterol. In the vehicle-treated control, the 3D volume of ovarian cancer cells increased, but the 2D area decreased because of cell aggregation. However, as ovarian cancer cells did not aggregate in response to stigmasterol treatment, the 3D volume decreased, and the cells spread laterally to increase the 2D area (Figure 1C,D). We investigated the cell states to confirm the occurrence of programmed cell apoptosis by stigmasterol based on the dot population of the upper-right quadrant. In the case of ES2 cells, the proportion of cells showing late apoptosis was increased by 1.9%, 7.8%, and 29.8% following treatment with 5, 10, and 20 μg/mL of stigmasterol, respectively, compared to the vehicle-treated controls, which showed a 1.1% increase in the proportion of cells showing late apoptosis (Figure 1E). In the case of OV90 cells, the proportion of cells in the upper-right quadrant was increased by 0.6%, 2.5%, and 8.5% following treatments with 5, 10, and 20 μg/mL of stigmasterol, respectively, compared to the vehicle-treated control, which showed a 0.1% increase in the proportion of cells in the upper-right quadrant (Figure 1F). In the evaluation of cell cycle progression, in the case of ES2 cells, the proportion of cells in the subG1 phase was increased by 11.1% following treatment with 20 μg/mL of stigmasterol (vehicle-treated controls showed a 0.8% increase) (Figure 1G). In the case of OV90 cells, the proportion of cells in the subG1 phase increased by 5.6% following treatment with 20 μg/mL of stigmasterol (vehicle-treated controls showed a 0.8% increase) (Figure 1H).

### 3.2. Changes in Mitochondrial Function and ROS Levels by Stigmasterol in ES2 and OV90 Cells

Mitochondrial function was dramatically altered by stigmasterol (0, 5, 10, and 20 μg/mL) in ES2 cells. Mitochondrial depolarization was increased to 700%, 1433%, and 3100% by 5, 10, and 20 μg/mL of stigmasterol, respectively, compared to that in the case of the vehicle-treated control (100%) in ES2 cells (Figure 2A). Similarly, the mitochondrial depolarization in OV90 cells was increased to 292%, 400%, and 492% by 5, 10, and 20 μg/mL of stigmasterol, respectively, compared to that in the case of the vehicle-treated control (100%) (Figure 2B). ROS production in ES2 cells was dose-dependently increased by 5.5%, 6.9%, and 10.4% by 5, 10, and 20 μg/mL of stigmasterol, respectively, compared to that in the case of the vehicle-treated control (5.1%) (Figure 2C). ROS generation in OV90 cells also increased by 3.8% and 4.1% by 10 and 20 μg/mL of stigmasterol, respectively, compared to that in the case of the vehicle-treated control (2.8%) (Figure 2D).

### 3.3. Upregulation of Cytosolic and Mitochondrial Calcium Levels by Stigmasterol in ES2 and OV90 Cells

To determine the regulation of calcium concentrations in the cytosol and mitochondria, we incubated ES2 and OV90 cells with 0, 5, 10, and 20 μg/mL of stigmasterol for 48 h. After this, the calcium levels of the cytosol and mitochondria were observed using Fluo-4-AM and Rhod-2 staining, respectively. The cytosolic calcium level of ES2 cells was increased by 4.1%, 5.8%, and 25.2% by 5, 10, and 20 μg/mL of stigmasterol, respectively, compared to that in the case of the vehicle-treated controls (3.9%) (Figure 3A). The cytosolic calcium level in OV90 cells was increased by 3.6%, 5.8%, and 6.6% by 5, 10, and 20 μg/mL of stigmasterol, respectively, compared to that in the case of the vehicle-treated control (3.5%) (Figure 3B). In addition, the mitochondrial calcium levels in ES2 cells were increased by 1.5%, 3.5%, and 4.0% by 5, 10, and 20 μg/mL of stigmasterol, respectively, compared to that in the case of the vehicle-treated control (1.1%) (Figure 3C). In OV90 cells, the mitochondrial calcium was upregulated by 5.8%, 8.9%, and 13.2% by 5, 10, and 20 μg/mL of stigmasterol, respectively, compared to that in the case of the vehicle-treated control (5.0%) (Figure 3D).

### 3.4. Regulation of Endoplasmic Reticulum Stress, the Endoplasmic Reticulum-Mitochondria Axis, and Autophagy by Stigmasterol in ES2 and OV90 Cells

Endoplasmic reticulum (ER) stress and the ER-mitochondria axis were investigated using Western blotting analysis. Activation of representative unfolded protein response (UPR) proteins was observed. Phosphorylated PKR-like ER resident kinase (p-PERK), phosphorylated eukaryotic translation-initiation factor 2α (p-eIF2α), inositol-requiring enzyme 1α (IRE1α), growth arrest and DNA damage-induced-153 (GADD153), activating transcription factor 6α (ATF6α), and glucose-regulated protein 78 (GRP78) were activated by stigmasterol (0, 5, 10, and 20 μg/mL) treatments, compared to the case for TUBA, in ES2 and OV90 cells. p-PERK level increased four point three-fold and two point six-fold in response to stigmasterol (20 μg/mL) treatment relative to that in vehicle-treated controls in ES2 and OV90 cells, respectively. p-eIF2α level increased six-fold and four point six-fold in response to stigmasterol treatment relative to that in vehicle-treated controls in ES2 and OV90 cells, respectively. IRE1α levels increased nine point nine-fold and seven point eight-fold in response to stigmasterol treatment relative to that in vehicle-treated controls in ES2 and OV90 cells, respectively. GADD153 expression increased three point nine-fold and six point two-fold in response to stigmasterol treatment relative to that in vehicle-treated controls in ES2 and OV90 cells, respectively. ATF6α expression increased three point six-fold and three point seven-fold in response to stigmasterol treatment relative to that in controls in ES2 and OV90 cells, respectively. GRP78 expression was upregulated two-fold and three-fold in response to stigmasterol treatment relative to that in vehicle-treated controls in ES2 and OV90 cells, respectively (Figure 4A,B). In addition, voltage-dependent anion channel (VDAC), inositol 1,4,5-triphosphate receptor 1 (IP3R1), IP3R2, vesicle-associated membrane protein-associated protein B/C (VAPB), family with sequence similarity 82, member a2 (FAM82A2), beclin-1 (BECN1), phosphorylated UNC-51-like kinase 1 (p-ULK1), and autophagy-related 5 (ATG5) were activated in a dose-dependent manner by stigmasterol (0, 5, 10, and 20 μg/mL), compared to the case for TUBA, in ES2 and OV90 cells. VDAC expression increased two point six-fold and three point seven-fold in response to stigmasterol (20 μg/mL) treatment relative to that in vehicle-treated controls in ES2 and OV90 cells, respectively. IP3RI expression was upregulated three point seven-fold and three-fold upon stigmasterol treatment relative to that in vehicle-treated controls in ES2 and OV90 cells, respectively. IP3R2 expression increased three point eight-fold and one point six-fold upon stigmasterol treatment in ES2 and OV90 cells, respectively. VAPB expression increased three point seven-fold and one point eight-fold upon stigmasterol treatment relative to that in vehicle-treated controls in ES2 and OV90 cells, respectively. FAM82A2 expression increased five point two-fold and three point four-fold upon stigmasterol treatment relative to that in vehicle-treated controls in ES2 and OV90 cells, respectively. p-ULK1 level increased two point four-fold and two point eight-fold upon stigmasterol treatment relative to that in vehicle-treated controls in ES2 and OV90 cells, respectively. BECN1 was upregulated four point two-fold and three point four-fold upon stigmasterol treatment relative to that in vehicle-treated controls in ES2 and OV90 cells, respectively. ATG5 expression increased two point four-fold and two point three-fold upon stigmasterol treatment relative to that in vehicle-treated controls in ES2 and OV90 cells, respectively (Figure 4C–F).

### 3.5. Restriction of Growth and Inactivation of Intracellular Signals by Stigmasterol in ES2 and OV90 Cells

Stigmasterol inhibited cell growth in both ES2 and OV90 cells. The cell growth of ES2 cells was decreased by up to 50.0% when treated with 20 μg/mL of stigmasterol (Figure 5A). The cell proliferation of OV90 cells was restricted by up to 54.3% when treated with 20 μg/mL of stigmasterol (Figure 5B). Additionally, the phosphorylation of AKT, P70S6K, S6, ERK1/2, JNK, and P38 was inhibited by stigmasterol in a dose-dependent manner (Figure 5C,D). Next, we used inhibitors to verify the correlation of stigmasterol with the intracellular signaling mechanism. We used the PI3K inhibitor (LY294002; 20 μM), ERK1/2 inhibitor (U0126; 20 μM), JNK inhibitor (SP600125; 20 μM), and P38 inhibitor (SB203580; 20 μM) with stigmasterol. ES2 and OV90 cells were treated with these inhibitors prior to stigmasterol treatment (Figure 6). Phosphorylation of AKT in ES2 cells was almost completely inhibited by all these inhibitors but was only inhibited by LY294002 and SB203580 in OV90 cells. The phosphorylation of P70S6K was almost completely inactivated by LY294002, U0126, and SB203580 in ES2 cells and by LY294002, SP600125, and SB203580 in OV90 cells. The phosphorylation of S6 protein was almost completely blocked by all inhibitors except U0126 in ES2 cells and was completely blocked by all inhibitors except SP600125 in OV90 cells. The phosphorylation of ERK1/2 was completely blocked by U0126 and SB203580 in both cell types and only slightly blocked by LY294002 in ES2 cells. JNK phosphorylation was completely blocked by all inhibitors in both cells. The phosphorylation of P38 was completely blocked by all inhibitors except SP600125 in both cell lines.

### 3.6. Calcium Regulation by Stigmasterol in the Death of ES2 and OV90 Cells

Ruthenium red, an inhibitor of calcium overload, inhibited cell apoptosis induced by stigmasterol in ES2 and OV90 cells. Additional treatment of ruthenium red to stigmasterol decreased cell apoptosis from 9.3% (stigmasterol alone) to 3.6% and from 7.7% to 3.4% in ES2 and OV90 cells, respectively (Figure 7A,B). Further, the stigmasterol-increased mitochondrial depolarization was decreased from 7.0% to 2.1% and from 5.9% to 2.6% in ES2 and OV90 cells, respectively, following cotreatment with stigmasterol and ruthenium red (Figure 7C,D). Mitochondrial calcium levels were increased by stigmasterol by up to 8.9% and 6.9% in ES2 and OV90 cells, respectively. However, treatment with stigmasterol and ruthenium red reduced the mitochondrial calcium from 8.9% to 3.8% and from 6.9% to 3.7% in ES2 and OV90 cells, respectively (Figure 7E,F). 

### 3.7. Decreased Migration Activity of ES2 and OV90 Cells Following Stigmasterol Treatment

Cell migration was inhibited by stigmasterol treatment, decreasing to 28.2% in ES2 cells and to 27.4% in OV90 cells compared to the respective vehicle-treated control cells (Figure 8A,B). Expression of genes that played a pivotal role in cancer cell migration, including *VEGFA*, *PLAU*, *MMP2*, *MMP9*, and *MMP14*, were reduced in dose-dependent manner of stigmasterol in both cell lines (Figure 8C,D).

## 4. Discussion

The pharmacological functions of phytosterol, a plant-derived natural product, have been highly studied. Stigmasterol, a type of phytosterol, has been reported to have various physiological effects [4]. Recently, stigmasterol has been shown to have anticancer effects against various cancers. However, the anticancer effects of stigmasterol on ovarian cancer have not yet been reported. Stigmasterol suppresses skin cancer by increasing the lipid peroxide levels and causing DNA damage [16] and inhibits the development of cholangiocarcinoma via the downregulation of TNF-alpha and VEGFR-2 [14]. Stigmasterol represses the development of prostate cancer by inducing p53 protein expression, while inhibiting p21 and p27 protein expression [20] and induces an increase of proapoptotic signals (BAX and p53) while decreasing antiapoptotic signals (Bcl-2) in liver cancer. Further, stigmasterol promotes the mitochondrial apoptosis signaling pathway, including the upregulation of caspase-8 and -9, in hepatoma. Additionally, stigmasterol induces DNA damage and increases apoptosis in HepG2 cells [13]. Similarly, our results illustrated that stigmasterol induced the activity of apoptotic proteins, including cleaved caspase-3, cleaved caspase-9, cytochrome *c*, BAK, and BAX.

Mitochondria are important organelles that control cell survival [25]. Mitochondria regulate the cleavage of caspases, production of cytochrome c, and activation of BAX and BAK to induce programmed cell apoptosis [26]. Moreover, increased mitochondrial calcium levels can promote the activation of proapoptotic factors via destruction of the mitochondria. This causes an increase in calcium concentration in the cytosol and induces cell apoptosis [27]. Increased calcium levels in the cytosol stimulate proapoptotic signals, while repressing antiapoptotic signals [28]. Cisplatin increases ROS in cancer cells, leading to irreversible cell death. Excessive ROS production activates antitumor signals and programmed cell death [29]. In our study, stigmasterol induced the apoptosis of ovarian cancer cells by causing mitochondrial malfunction, ROS production, and calcium overload in the mitochondria and cytosol. In addition, the subG1 phage was significantly increased upon stigmasterol treatment in ovarian cancer cells. Ruthenium red prevents mitochondrial calcium overload [30]. The cotreatment of stigmasterol and ruthenium red reduced mitochondrial depolarization and cell death, compared to treatment with stigmasterol alone. These results suggest that stigmasterol-induced cell apoptosis occurs via the accumulation of mitochondrial calcium in ovarian cancer cells.

The ER is an important organelle that controls protein translocation, folding, and post-transcriptional modifications in eukaryotic cells. The accumulation of ER stress can induce cancer cell death [31]. Stigmasterol activated ER stress sensor proteins and ER-mitochondria axis proteins in ovarian cancer cells, indicating that stigmasterol induces anticancer effects mediated by the regulation of the ER-mitochondria axis. Additionally, stigmasterol decreased cell proliferation and inhibited cell cycle progression in ES3 and OV90 cells. The PI3K and MAPK signal cascades are pivotal in cancer cell proliferation and cell cycle progression [32]. In ovarian cancer, MAPK/PI3K signals are frequently activated. PI3K pathway inhibition may be the most useful in combination with the MAPK suppressor in ovarian cancer patients [33] Therefore, the inhibition of PI3K/MAPK signal cascades is useful for the treatment of ovarian cancer. The spheroid model provides a good opportunity to confirm the clinical therapeutic effectiveness of various drugs due to their association with the malignant nature of cancer cells, such as tumorigenicity or chemoresistance [34]. Stigmasterol effectively inhibited the aggregation of ovarian cancer cells. Ovarian cancer cells failed to form tumors and were scattered. 

*VEGFA* stimulates cell mitogenesis and cell migration in ovarian cancer cells [35]. *PLAU* induces migration and metastasis in breast cancer [36]. Metalloproteinases (MMPs) are overexpressed in several tumor environments and can promote metastasis and migration for cancer development [37]. In ES2 and OV90 cells, *VEGFA*, *PLAU*, *MMP2*, *MMP9*, and *MMP14* expression levels were reduced by stigmasterol treatments. Taken together, we are the first to show that stigmasterol exerts a complex anticancer effect in the context of ovarian cancer. This effect can improve the anticancer effect of standard ovarian cancer drugs used to treat recurrent ovarian cancer. Therefore, these findings suggest that stigmasterol can be used as a new treatment option for ovarian cancer.

## 5. Conclusions

Collectively, our results suggest that stigmasterol dose-dependently induced cell apoptotic signals in ES2 and OV90 cells. It also caused an increase in mitochondrial depolarization and ROS generation and induced abnormal increases in calcium levels in the cytosol and mitochondria. ER stress sensor protein levels were increased by stigmasterol (Figure 9). As a result, cell death occurred in ovarian cancer cell lines treated with stigmasterol. Additionally, stigmasterol inhibited cell growth by controlling cell proliferation and the cell cycle in the tested ovarian cancer cell lines. Stigmasterol also reduced cell migration and inhibited cell growth-related signaling cascades (e.g., PI3K/MAPK) in ovarian cancer cells. This evidence suggests that stigmasterol could be used as a new treatment modality for ovarian cancer.

## Figures and Tables

**Figure 1 pharmaceutics-12-00488-f001:**
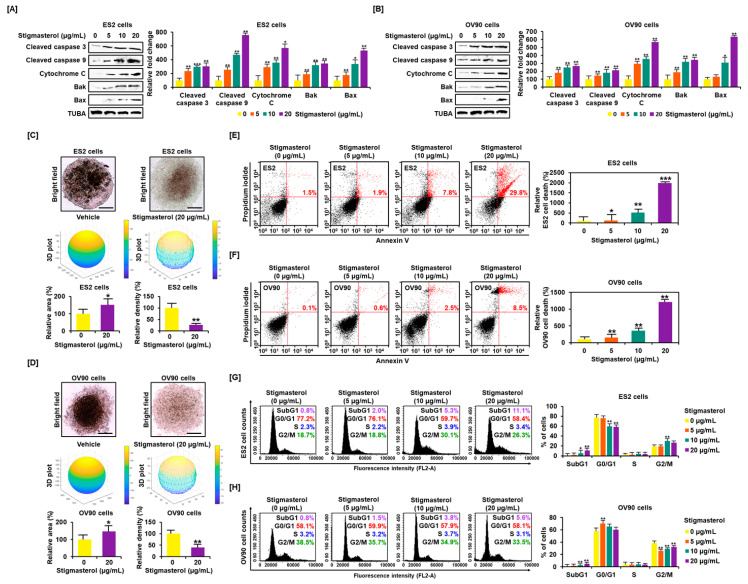
Stigmasterol affects ovarian cancer cell apoptosis and tumor formation in ES2 and OV90 cells. (**A**,**B**) Western blot bands showed the expression of proapoptotic signaling molecules in both cell types following stigmasterol treatments (0, 5, 10, and 20 μg/mL). Alpha-tubulin (TUBA) was used as a control. (**C**,**D**) Spheroid formation of ovarian cancer cells was compared between vehicle-treated cells and stigmasterol-treated cells. (**E**,**F**) Annexin V and propidium iodide (PI) staining were performed to determine cell death in ES2 and OV90 cells. The quadrant of the dot blot represents the state of apoptosis in ES2 and OV90 cells. The comparative graph represents changes in late apoptosis due to stigmasterol treatment (0, 5, 10, and 20 μg/mL) compared to the vehicle-treated control (100%) in ES2 and OV90 cells. (**G**,**H**) Histogram presents cell cycle progression in stigmasterol-treated (0, 5, 10, and 20 μg/mL) ovarian cancer cells. Comparative graph represents the % of cells in the subG1, G0/G1, S, and G2/M phases in stigmasterol-treated (0, 5, 10, and 20 μg/mL) ovarian cancer cells.

**Figure 2 pharmaceutics-12-00488-f002:**
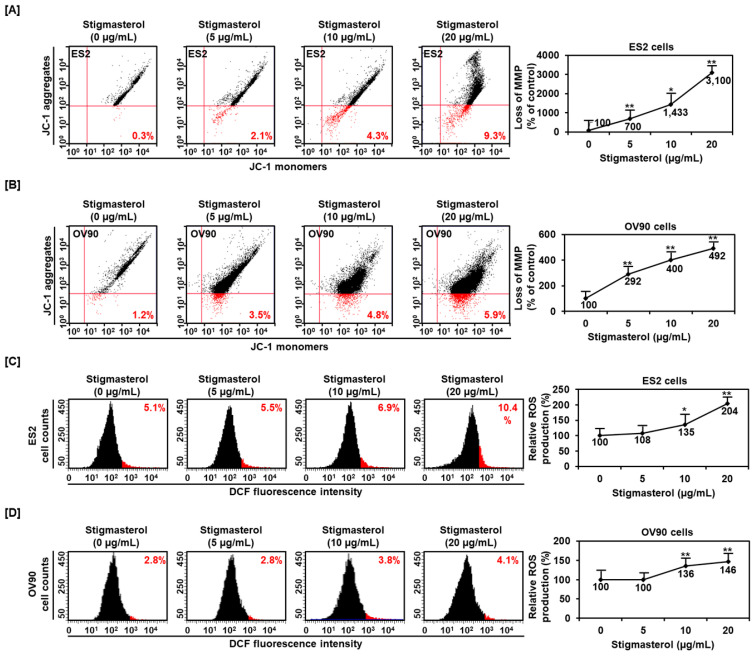
Alteration of mitochondrial membrane potential and ROS generation by stigmasterol in both cells. (**A**,**B**) JC-1 dye was used to investigate the alteration of mitochondrial membrane potential by stigmasterol (0, 5, 10, and 20 μg/mL). Quadrant of the dot blot represented the state of mitochondrial membrane potential in ES2 and OV90 cells. Comparative graph represented the loss of the mitochondrial membrane potential compared to the control group (100%) in ES2 and OV90 cells. (**C**,**D**) DCF fluorescence intensity was observed to investigate the change by stigmasterol treatment (0, 5, 10, and 20 μg/mL). The histogram represented the state of ROS production in ES2 (Figure 2C, left) and OV90 (Figure 2D, left) cells. Comparative graph represented ROS production compared to vehicle-treated control (100%) in ES2 (Figure 2C, right) and OV90 (Figure 2D, right) cells. DCF: dichlorofluorescein.

**Figure 3 pharmaceutics-12-00488-f003:**
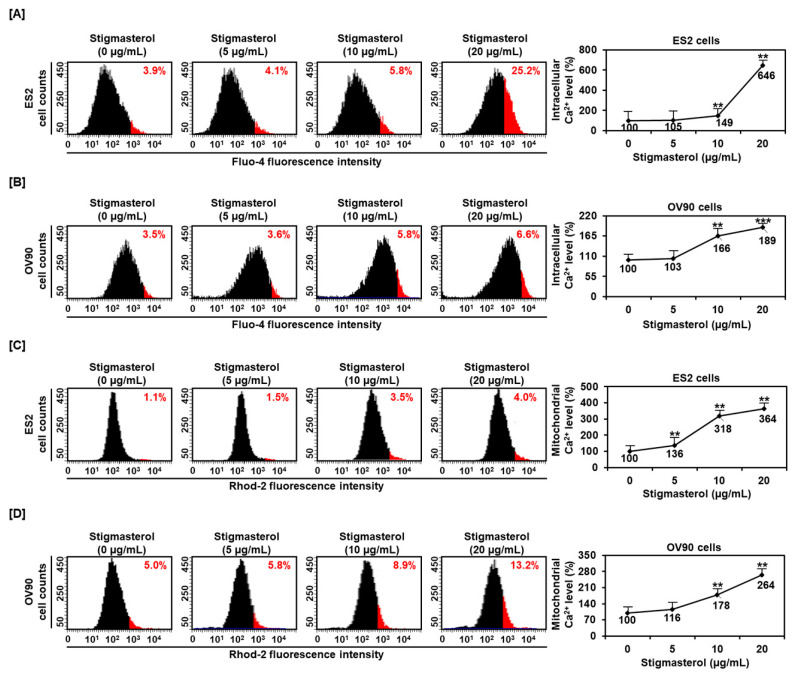
Calcium concentrations in the cytosol and mitochondria in stigmasterol-treated ovarian cancer cells. (**A**,**B**) Fluo-4 dye was used to determine the intracellular calcium levels. The histogram represents the change of cytosolic calcium levels following stigmasterol treatments (0, 5, 10, and 20 μg/mL) in ES2 (Figure 3A, left) and OV90 (Figure 3B, left) cells. The comparative graph represents changes in intracellular calcium levels following stigmasterol treatments compared to the vehicle-treated controls (100%) in ES2 (Figure 3A, right) and OV90 (Figure 3B, right) cells. (**C**,**D**) Rhod-2 fluorescence was used to determine the mitochondrial calcium levels in stigmasterol-treated ovarian cancer cells. The histogram represents changes in mitochondrial calcium levels following stigmasterol treatments (0, 5, 10, and 20 μg/mL) in ES2 (Figure 3C, left) and OV90 (Figure 3D, left) cells. The comparative graph represents changes in mitochondrial calcium levels following stigmasterol treatments compared to the vehicle-treated control (100%) in ES2 (Figure 3C, right) and OV90 (Figure 3D, right) cells.

**Figure 4 pharmaceutics-12-00488-f004:**
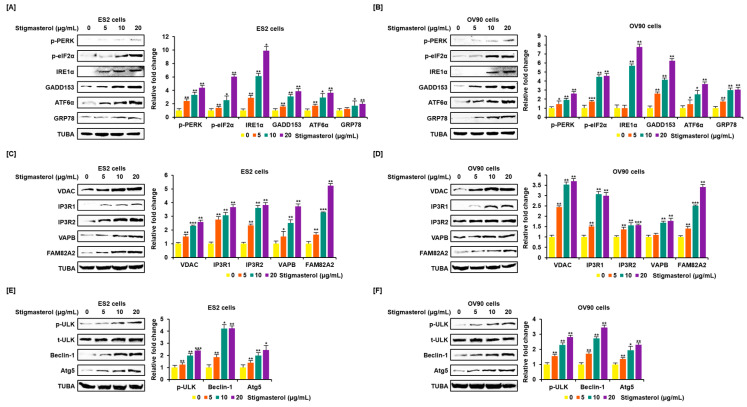
Increased expression of endoplasmic reticulum (ER) stress sensor proteins and ER-mitochondrial axis proteins due to stigmasterol treatment in ovarian cancer cells. (**A**,**B**) Immunoblots representing the expression of unfolded protein response (UPR) proteins following stigmasterol treatments (0, 5, 10, and 20 μg/mL). TUBA was used as a control and is shown at the bottom of the figure. (**C**,**D**) Immunoblots representing the expression of ER-mitochondrial axis proteins following stigmasterol treatments (0, 5, 10, and 20 μg/mL). TUBA was used as a control and is shown at the bottom of the figure. (**E**,**F**) Immunoblots representing the expression of autophagy proteins following stigmasterol treatment. TUBA was used as a control and is shown at the bottom of the figure.

**Figure 5 pharmaceutics-12-00488-f005:**
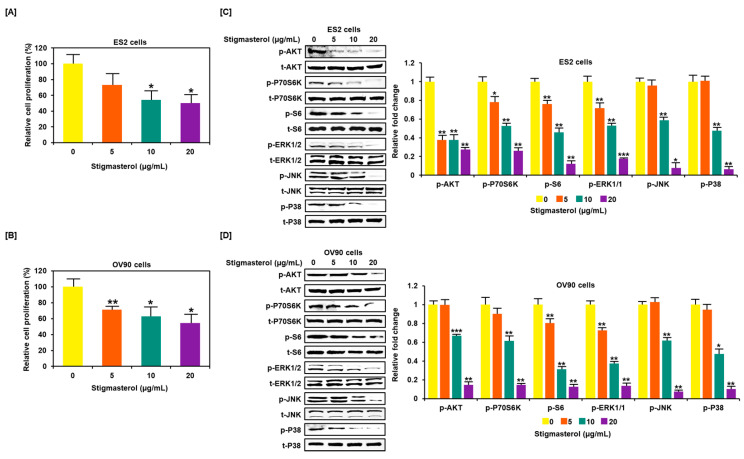
Restriction of cell growth and cell growth-related signals in ovarian cancer cells following stigmasterol treatments. (**A**,**B**) Cell proliferation data shows the cell viability (%) following the treatment of ovarian cancer cells with stigmasterol. The comparative graph represents the relative proliferation of each cell type following treatments with stigmasterol, compared to vehicle-treated control cells (100%). (**C**,**D**) Western blots present the changes of PI3K and MAPK expression following the treatments of ovarian cancer cells with stigmasterol (0, 5, 10, and 20 μg/mL).

**Figure 6 pharmaceutics-12-00488-f006:**
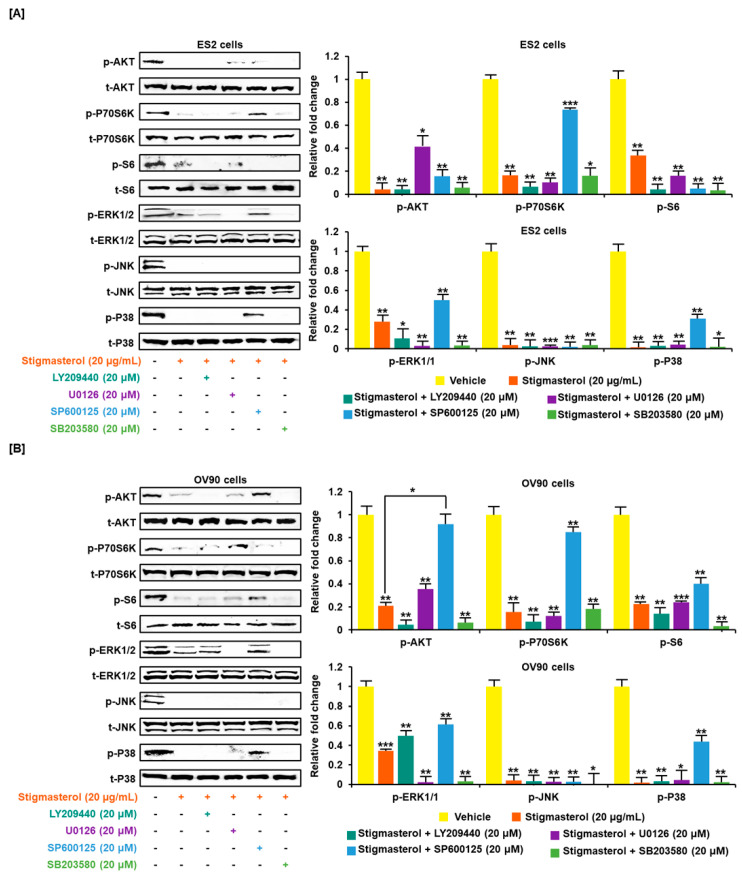
Cotreatment of stigmasterol and inhibitors in ovarian cancer cells. (**A**,**B**) Western blots showed changes of PI3K and MAPK by cotreatment with stigmasterol or each inhibitor in ovarian cancer cells. Comparative graph represented the change of phosphorylation compared to the vehicle-treated control (100%) in ES2 and OV90 cells.

**Figure 7 pharmaceutics-12-00488-f007:**
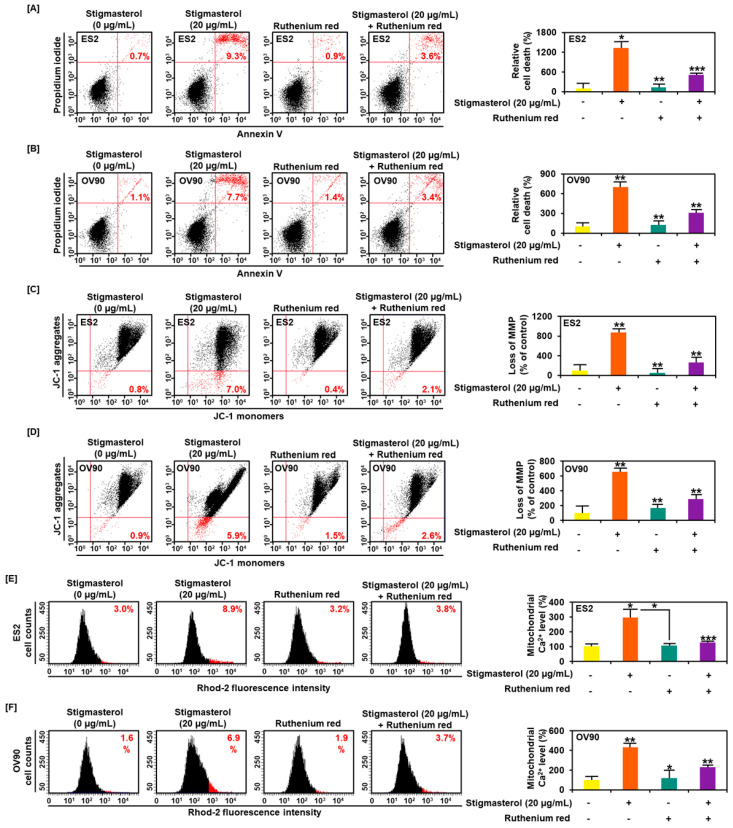
Calcium regulation and cell apoptosis by stigmasterol in ovarian cancer. (**A**,**B**) Annexin V and PI were observed to investigate the cell death of ES2 and OV90 cells. Quadrant of the dot blot represented the state of cell apoptosis in ES2 and OV90 cells. Comparative graph represented the change of late apoptosis by stigmasterol or a combination with ruthenium red compared to the control (100%) in ES2 and OV90 cells. (**C**,**D**) JC-1 dye was used to investigate the alteration of the mitochondrial membrane potential by stigmasterol or a combination with ruthenium red. Quadrant of the dot blot represented the state of the mitochondrial membrane potential. Comparative graph represented the loss of the mitochondrial membrane potential compared to the control group (100%). (**E**,**F**) Rhod-2 fluorescence was observed to investigate the mitochondrial calcium levels in stigmasterol or a combination with ruthenium red. The histogram represented a change of the mitochondrial calcium. Comparative graph represented the change of the mitochondrial calcium levels.

**Figure 8 pharmaceutics-12-00488-f008:**
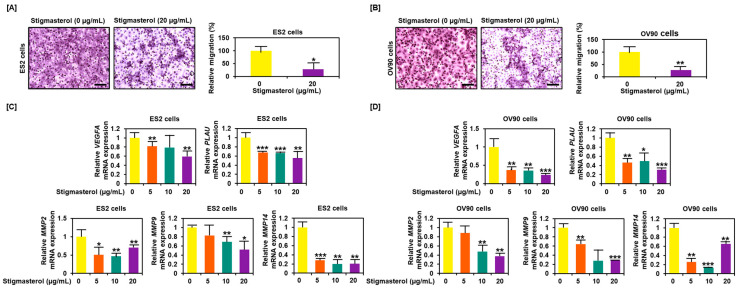
Reduced cell migration by stigmasterol in ovarian cancer cells. (**A**,**B**) Cell migration was observed using Transwell inserts. Migrated cells through the membrane were measured on images of five nonoverlapping locations. (**C**,**D**) Gene expression of migration-related genes were investigated by quantitative real-time (RT)-PCR. All experiments were conducted in triplicate. Scale bar represents 100 μm.

**Figure 9 pharmaceutics-12-00488-f009:**
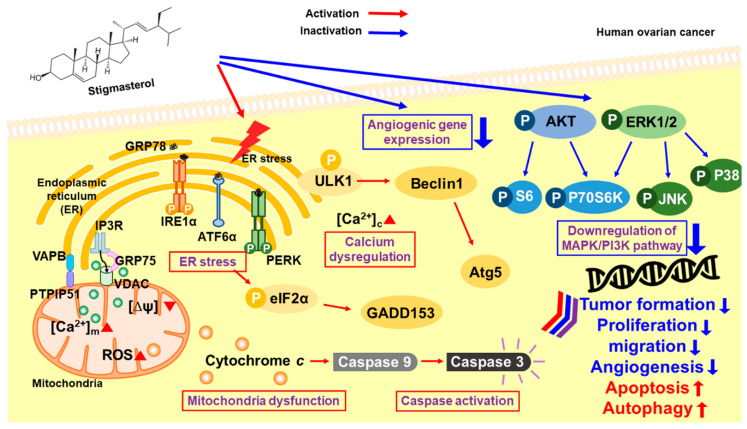
Schematic illustration of the mechanisms downstream of stigmasterol in human ovarian cancer cells.

**Table 1 pharmaceutics-12-00488-t001:** List of antibodies.

Primary Antibodies	Dilution	Supplier	Catalog Number
Cleaved caspase-3	1:1000	Cell Signaling	9664
Cleaved caspase-9	1:1000	Cell Signaling	9501
Cytochrome c	1:1000	Cell Signaling	11940
BAK	1:1000	Cell Signaling	12105
BAX	1:1000	Cell Signaling	2774
Phospho-PERK (Thr^981^)	1:1000	Santa Cruz	sc-32577
Phospho-eIF2α (Ser^51^)	1:1000	Cell Signaling	3398
IRE1α	1:1000	Cell Signaling	3294
GADD153	1:1000	Santa Cruz	sc-7351
ATF6α	1:1000	Santa Cruz	sc-166659
GRP78	1:1000	Santa Cruz	sc-13968
TUBA	1:1000	Santa Cruz	sc-5286
VDAC	1:1000	Cell Signaling	4661
IP3R1	1:1000	Invitrogen	PA1-901
IP3R2	1:1000	Santacruz	sc-398434
VAPB	1:1000	Invitrogen	PA5-53023
FAM82A2	1:1000	Abcam	ab182105
Phospho-ULK1 (SER^555^)	1:1000	Cell Signaling	5869
ULK1	1:1000	Cell Signaling	8054
BECN1	1:1000	Cell Signaling	3495
ATG5	1:1000	Cell Signaling	12994
Phospho-AKT (SER^473^)	1:1000	Cell Signaling	4060
AKT	1:1000	Cell Signaling	9272
Phospho-P70S6K (Thr^421^/Ser^424^)	1:1000	Cell Signaling	9204
P70S6K	1:1000	Cell Signaling	9202
Phospho-S6 (Ser^235/236^)	1:1000	Cell Signaling	2211
S6	1:1000	Cell Signaling	2217
Phospho-ERK1/2(Thr^202^/Tyr^204^)	1:1000	Cell Signaling	9101
ERK1/2	1:1000	Cell Signaling	4695
Phospho-JNK (Thr^183^/Tyr^185^)	1:1000	Cell Signaling	4668
JNK	1:1000	Cell Signaling	9252
Phospho-P38 (Thr^180^/Tyr^182^)	1:1000	Cell Signaling	4511
P38	1:1000	Cell Signaling	9212

**Table 2 pharmaceutics-12-00488-t002:** List of primers.

Primer	Sequence		Size
*VEGFA*	Forward	5′-TTGTACAAGATCCGCAGACG-3′	100 bp
	Reverse	5′-TCACATCTGCAAGTACGTTCG-3′	
*PLAU*	Forward	5′-CAACTGCCCAAAGAAATTCG-3′	148 bp
	Reverse	5′-AAGGACAGTGGCAGAGTTCC-3′	
*MMP2*	Forward	5′-GGCATTCAGGAGCTCTATGG-3′	137 bp
	Reverse	5′-ATCTCACCACGGATCTGAGC-3′	
*MMP9*	Forward	5′-TTGACAGCGACAAGAAGTGG-3′	145 bp
	Reverse	5′-TCAGTGAAGCGGTACATAGGG-3′	
*MMP14*	Forward	5′-GCAGAAGTTTTACGGCTTGC-3′	117 bp
	Reverse	5′-ACATTGGCCTTGATCTCAGC-3′	

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
