# Peer review of "Stigmasterol Causes Ovarian Cancer Cell Apoptosis by Inducing Endoplasmic Reticulum and Mitochondrial Dysfunction"

_pharmaceutics, 2020, doi:10.3390/pharmaceutics12060488_

Round 1
Reviewer 1 Report
In this study, the authors investigated the role of stigmasterol in ovarian and tried to demonstrate a panel of related markers of endoplasmic reticulum stress, endoplasmic reticulum-mitochondria axis and autophagy. Overall, the experimental design of this study is okay. However, there are many questions need to be addressed before it could further consider for publication. Below are the comments/suggestions:
- Line 4: The dot in title should be removed.
- Line167-169: please explain why the tumor area increased but the tumor volume decreased?
- In Result 3.4, please shown the transcript level of these targets? Could the authors explain the potential receptor of stigmasterol in ovarian cancer?
- The authors also mentioned anticancer function of stigmasterol in other cancer type. Is it the same in other cancer type? Please include more info in the discussion section. I think it was be better if the authors could show the advantage of stigmasterol in ovarian cancer when comparing with other cancer type. Otherwise, this study would not have much innovation.
- In both 3.5 and 3.6, the authors are talking about the inactivation of intracellular signals by stigmasterol in ES2 and OV90 cells, I think these two sections better to be combined.
- Actually, what is the main point for 3.6? Are the authors trying to study the potential proteins targeted by stigmasterol? Or where is the potential target location for the intracellular signaling? I did not get the point there.
- A large number of data were shown in results section, but it is unconvincing when discussing the effects of stigmasterol in ovarian cancer. More explanation should be providing to illustrate the PI3K/MAPK signal pathway.
Author Response
Reviewer 1
In this study, the authors investigated the role of stigmasterol in ovarian and tried to demonstrate a panel of related markers of endoplasmic reticulum stress, endoplasmic reticulum-mitochondria axis and autophagy. Overall, the experimental design of this study is okay. However, there are many questions need to be addressed before it could further consider for publication. Below are the comments/suggestions:
Response: We appreciate the reviewer’s valuable comments and suggestions on our manuscript. We have substantially revised our manuscript accordingly. To address the reviewer’s comments, we have prepared a point-by-point response to each comment and highlighted the changes in yellow.
- Line 4: The dot in title should be removed.
Response: As reviewers commented, we have removed the dot in title.
- Line167-169: please explain why the tumor area increased but the tumor volume decreased?
Response: We identified that ovarian cancer cells cannot aggregate in the presence of stigmasterol. In the vehicle-treated control, the 3D volume of ovarian cancer cells increased, but the 2D area decreased because of cell aggregation. However, as ovarian cancer cells did not aggregate in response to stigmasterol treatment, the 3D volume decreased, and the cells spread laterally to increase the 2D area.
- In Result 3.4, please shown the transcript level of these targets? Could the authors explain the potential receptor of stigmasterol in ovarian cancer?
Response: We revised the manuscript and added transcript levels of targets in the result section 3.4. We did not deal with the receptor for stigmasterol but we think that stigmasterol activated some receptors in ES2 and OV90 cells. The effects of stigmasterol on bile acid nuclear receptor FXR [1], liver X Receptor [2], epidermal growth factor receptor [3] have been reported. However, the receptors acted on by stigmasterol have not been fully understood and should be studied in the future.
- The authors also mentioned anticancer function of stigmasterol in other cancer type. Is it the same in other cancer type? Please include more info in the discussion section. I think it was be better if the authors could show the advantage of stigmasterol in ovarian cancer when comparing with other cancer type. Otherwise, this study would not have much innovation.
Response: Stigmasterol inhibits the development of skin cancer by increasing lipid peroxide levels and causing DNA damage [4] and inhibits the development of cholangiocarcinoma via the downregulation of TNF-alpha and VEGFR-2 [5]. It also inhibits the development of prostate cancer by inducing p53 expression, while inhibiting p21 and p27 expression [6]. Additionally, stigmasterol induces an increase in pro-apoptotic signals (BAX, p53), while decreasing the anti-apoptotic signals (Bcl-2) in liver cancer. Further, stigmasterol promotes the mitochondrial apoptotic signaling pathway; this includes upregulation of caspases-8 and -9, in hepatoma. Additionally, stigmasterol induces DNA damage and increases apoptosis in HepG2 cells [7]. Similarly, our results illustrated that stigmasterol induced the activity of apoptotic proteins including cleaved caspase-3, cleaved caspase-9, cytochrome c, BAK, and BAX. Taken together, we are the first to show that stigmasterol exerts a complex anti-cancer effect in the context of ovarian cancer. This effect can improve the anti-cancer effect of standard ovarian cancer drugs used to treat recurrent ovarian cancer.
- In both 3.5 and 3.6, the authors are talking about the inactivation of intracellular signals by stigmasterol in ES2 and OV90 cells, I think these two sections better to be combined.
Response: We combined the results sections 3.5 and 3.6.
- Actually, what is the main point for 3.6? Are the authors trying to study the potential proteins targeted by stigmasterol? Or where is the potential target location for the intracellular signaling? I did not get the point there.
Response: We identified changes in growth-related cell signals in stigmasterol-treated ovarian cancer cell lines. Stigmasterol downregulated MAPK and PI3K signaling cascades. Importantly, we identified cross-correlations in stigmasterol-mediated decreases between the signals.
- A large number of data were shown in results section, but it is unconvincing when discussing the effects of stigmasterol in ovarian cancer. More explanation should be providing to illustrate the PI3K/MAPK signal pathway.
Response: We have elaborated on the PI3K/MAPK signaling pathway in the discussion section.
[References]
- Carter, B.A.; Taylor, O.A.; Prendergast, D.R.; Zimmerman, T.L.; Von Furstenberg, R.; Moore, D.D.; Karpen, S.J. Stigmasterol, a soy lipid-derived phytosterol, is an antagonist of the bile acid nuclear receptor FXR. Pediatr Res 2007, 62, 301-306, doi:DOI 10.1203/PDR.0b013e3181256492.
- Marinozzi, M.; Navas, F.F.C.; Maggioni, D.; Carosati, E.; Bocci, G.; Carloncelli, M.; Giorgi, G.; Cruciani, G.; Fontana, R.; Russo, V. Side-Chain Modified Ergosterol and Stigmasterol Derivatives as Liver X Receptor Agonists. J Med Chem 2017, 60, 6548-6562, doi:10.1021/acs.jmedchem.7b00091.
- Nazemi, M.; Khaledi, M.; Golshan, M.; Ghorbani, M.; Amiran, M.R.; Darvishi, A.; Rahmanian, O. Cytotoxicity Activity and Druggability Studies of Sigmasterol Isolated from Marine Sponge Dysidea avara Against Oral Epithelial Cancer Cell (KB/C152) and T-Lymphocytic Leukemia Cell Line (Jurkat/ E6-1). Asian Pac J Cancer Prev 2020, 21, 997-1003, doi:10.31557/APJCP.2020.21.4.997.
- Ali, H.; Dixit, S.; Ali, D.; Alqahtani, S.M.; Alkahtani, S.; Alarifi, S. Isolation and evaluation of anticancer efficacy of stigmasterol in a mouse model of DMBA-induced skin carcinoma. Drug Des Devel Ther 2015, 9, 2793-2800, doi:10.2147/DDDT.S83514.
- Kangsamaksin, T.; Chaithongyot, S.; Wootthichairangsan, C.; Hanchaina, R.; Tangshewinsirikul, C.; Svasti, J. Lupeol and stigmasterol suppress tumor angiogenesis and inhibit cholangiocarcinoma growth in mice via downregulation of tumor necrosis factor-alpha. PLoS One 2017, 12, e0189628, doi:10.1371/journal.pone.0189628.
- Scholtysek, C.; Krukiewicz, A.A.; Alonso, J.L.; Sharma, K.P.; Sharma, P.C.; Goldmann, W.H. Characterizing components of the Saw Palmetto Berry Extract (SPBE) on prostate cancer cell growth and traction. Biochem Biophys Res Commun 2009, 379, 795-798, doi:10.1016/j.bbrc.2008.11.114.
- Kim, Y.S.; Li, X.F.; Kang, K.H.; Ryu, B.; Kim, S.K. Stigmasterol isolated from marine microalgae Navicula incerta induces apoptosis in human hepatoma HepG2 cells. BMB Rep 2014, 47, 433-438, doi:10.5483/bmbrep.2014.47.8.153.
Reviewer 2 Report
Authors study the anticancer effects of Stigmasterol against two human ovarian cancer cells in order to broaden the anticancer activity of this phytosterol. Here are some points raised:
1. Authors must give some details about the two cancer cell lines used
2. Figure 1 is missing, so all the corresponding data discussed are “on the air”
3. There are fold differences between the two cell lines concerning mitochondrial function and ROS production in Figure 2, and reversely, on the Ca++ levels in cytosol and mitochondria, in Figure 3. Can the authors make an assumption on that? Data from mitochondrial mass in the two cell lines, pro and after the Stigmasterol treatment, could give an explanation
4. Lanes 352-354: authors make an assumption for synergistic action with cisplatin, due to ROS production by both agents. This is an arbitrary conclusion and need experimental data
Author Response
Reviewer 2
Authors study the anticancer effects of Stigmasterol against two human ovarian cancer cells in order to broaden the anticancer activity of this phytosterol. Here are some points raised:
Response: We appreciate the reviewer’s valuable comments and suggestions on our manuscript. We have substantially revised our manuscript according to the reviewer’s suggestions. To address the reviewer’s comments, we prepared a point-by-point response to each comment of the reviewer and highlighted changes in text of the manuscript in yellow.
- Authors must give some details about the two cancer cell lines used
Response: We have provided details of the cell lines in the materials section.
- Figure 1 is missing, so all the corresponding data discussed are “on the air”
Response: We have added figure 1 to our manuscript.
- There are fold differences between the two cell lines concerning mitochondrial function and ROS production in Figure 2, and reversely, on the Ca++ levels in cytosol and mitochondria, in Figure 3. Can the authors make an assumption on that? Data from mitochondrial mass in the two cell lines, pro and after the Stigmasterol treatment, could give an explanation
Response: The results of assays regarding mitochondrial function, ROS production, and the Ca++ levels in cytosol and mitochondria were compared with vehicle-treated controls. Therefore, it is impossible to compare data between different experiments. Data already show the results pre- and post-stigmasterol.
- Lanes 352-354: authors make an assumption for synergistic action with cisplatin, due to ROS production by both agents. This is an arbitrary conclusion and need experimental data
Response: We deleted the lanes 352-354.
Reviewer 3 Report
Stigmasterol, an unsaturated plant sterol derived from various medicinal plants, has demonstrated tremendous pharmacological potential against a variety of diseases, including cancer. Here, in this manuscript, the authors investigate the potential of these compounds on ovarian cancer and understand its mechanism of activity in controlling the growth of this disease. The authors perform a series of studies that identify more than one inhibitory mechanism of Stigmasterol on ovarian cancer cells. The overall theme of the research could have clinical relevance. The manuscript is well written, except for some minor grammatical errors and missing technical details.
MINOR COMMENTS:
- Figure 1 is completely missing within the manuscript and I am not sure if this happened during the submission process.
- Since Figure 1 wasn’t available, it is not quite clear if stigmasterol triggered apoptosis and G2/M cell cycle arrest in ovarian cancer. If so, the authors should highlight this explicitly in the discussion section as an additional point.
- Since the manuscript aims to show the clinical potential of stigmasterol in treating ovarian cancer, it would be interesting to know if there is an effect on the JAK/STAT signaling pathway. Upregulation of OCT4, a cancer stem-cell marker is known to activate the JAK/STAT signaling pathway which in turn increases viability, cancer cell invasion, tumorigenesis and enhances drug resistance. Perhaps on additional study highlighting the activation levels of JAK/STAT and OCT4 could improve the manuscripts clinical impact further.
- Please include a graphical abstract demonstrating the overall scheme of pathways/mechanism studied and identified.
Author Response
Reviewer 3
Stigmasterol, an unsaturated plant sterol derived from various medicinal plants, has demonstrated tremendous pharmacological potential against a variety of diseases, including cancer. Here, in this manuscript, the authors investigate the potential of these compounds on ovarian cancer and understand its mechanism of activity in controlling the growth of this disease. The authors perform a series of studies that identify more than one inhibitory mechanism of Stigmasterol on ovarian cancer cells. The overall theme of the research could have clinical relevance. The manuscript is well written, except for some minor grammatical errors and missing technical details.
Response: We appreciate the reviewer’s valuable comments and suggestions on our manuscript. We have substantially revised our manuscript according to the reviewer’s suggestions. To address the reviewer’s comments, we prepared a point-by-point response to each comment of the reviewer and highlighted changes in text of the manuscript in yellow.
MINOR COMMENTS:
- Figure 1 is completely missing within the manuscript and I am not sure if this happened during the submission process.
Response: We have added figure 1 to our manuscript.
- Since Figure 1 wasn’t available, it is not quite clear if stigmasterol triggered apoptosis and G2/M cell cycle arrest in ovarian cancer. If so, the authors should highlight this explicitly in the discussion section as an additional point.
Response: We have added figure 1 to our manuscript. Importantly, we have also added information about cell cycle progression in the discussion section.
- Since the manuscript aims to show the clinical potential of stigmasterol in treating ovarian cancer, it would be interesting to know if there is an effect on the JAK/STAT signaling pathway. Upregulation of OCT4, a cancer stem-cell marker is known to activate the JAK/STAT signaling pathway which in turn increases viability, cancer cell invasion, tumorigenesis and enhances drug resistance. Perhaps on additional study highlighting the activation levels of JAK/STAT and OCT4 could improve the manuscripts clinical impact further.
Response: As stigmasterol is known to inhibit the JAK/STAT signaling pathway in human gastric cancer cells [1], the reviewer’s suggestion is very interesting and valuable. However, we do not have access to reagents like antibodies. Recently, due to the corona pandemic, it is difficult to procure reagents from other countries.
- Please include a graphical abstract demonstrating the overall scheme of pathways/mechanism studied and identified.
Response: We appreciate the reviewer’s valuable comments and suggestions regarding our manuscript. We have added the schematic illustration (Figure 9.)
[References]
- Li, K.; Yuan, D.; Yan, R.; Meng, L.; Zhang, Y.; Zhu, K. Stigmasterol exhibits potent antitumor effects in human gastric cancer cells mediated via inhibition of cell migration, cell cycle arrest, mitochondrial mediated apoptosis and inhibition of JAK/STAT signalling pathway. J BUON 2018, 23
Round 2
Reviewer 1 Report
Thank you for your revised manuscript. I have no further comments.
Just one thing, for some reasons, there was a kind of error in Reference 1. Please kindly clarify.
Response: We appreciate the reviewer’s valuable comment on our manuscript. We have carefully revised our manuscript according to the reviewer’s suggestions. Thank you very much again.
Reviewer 2 Report
Authors improved their manuscript. it could be better.
Response: We appreciate the reviewer’s valuable comment and encouragement on our manuscript.